# Angiomatoid Fibrous Histiocytoma (AFH) of the Right Arm: An Exceptional Case with Pulmonary Metastasis and Confirmatory EWSR1::CREB1 Translocation

**DOI:** 10.3390/diagnostics12112616

**Published:** 2022-10-28

**Authors:** Gerardo Cazzato, Carmelo Lupo, Nadia Casatta, Flavia Riefoli, Andrea Marzullo, Anna Colagrande, Eliano Cascardi, Senia Maria Rosaria Trabucco, Giuseppe Ingravallo, Biagio Moretti, Eugenio Maiorano, Vito Pesce, Leonardo Resta

**Affiliations:** 1Section of Molecular Pathology, Department of Emergency and Organ Transplantation (DETO), University of Bari “Aldo Moro”, 70124 Bari, Italy; 2Innovation Department, Diapath S.P.A., Via Savoldini n.71, 24057 Martinengo, Italy; 3Section of Orthopedics and Traumatology, Department of Basic Medical Sciences, Neuroscience and Sense Organs, University of Bari “Aldo Moro”, 70124 Bari, Italy; 4Department of Medical Sciences, University of Turin, 10124 Turin, Italy; 5Pathology Unit, FPO-IRCCS Candiolo Cancer Institute, Str. Provinciale 142, km 3.95, 10060 Candiolo, Italy; 6Section of Orthopedics and Traumatology, Department of Clinical and Experimental Medicine, University of Foggia, 71100 Foggia, Italy

**Keywords:** angiomatoid fibrous histiocytoma, AFH, pulmonary metastasis, malignant, soft tissue, differential diagnosis

## Abstract

Angiomatoid fibrous histiocytoma (AFH) is a rare neoplasm described for the first time by Enzinger in 1979, and classified by World Health Organization 2020 as intermediate malignant potential neoplasm. It mostly occurs in the subcutis and is characterized by varying proportions of epithelioid, ovoid and spindle cells in a nodular and syncytial growth pattern, with some hemorrhagic pseudovascular spaces. In this paper, we report the clinical case of a 62-year-old man who presented with AFH on the right arm, and relapsed three years after first surgical excision. After a further three years, the patient presented with an intramuscular localization of AFH, and 12 months after this, a pulmonary metastasis of AFH was diagnosed. Given the rarity of the spreading of AFH, we performed Fluorescence In Situ Hybridization (FISH) and we detected EWSR1::CREB1 gene fusion.

**Figure 1 diagnostics-12-02616-f001:**
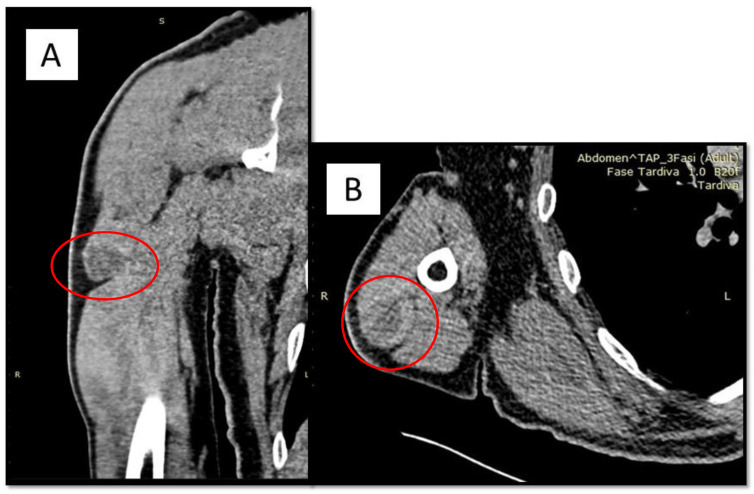
(**A**,**B**) CT findings of oval formation (red circle), 36 × 27 mm of diameter, in the lateral belly of the triceps brachial, located in the proximal third of the muscle, capsulated and well-defined in the surrounding muscle, except for minimal irregularities of the antero-lateral side. A 62-year-old man presented himself to the referral doctor in 2018 for unspecified pain in his left arm. After an orthopedic specialist consultation, it was decided to perform a CT scan with and without contrast medium in an attempt to localize the lesion. The medical history was silent for other pathologies and/or neoplasms. Both total body CT and regional Magnetic Resonance highlighted the presence of an oval formation in the lateral belly of the brachial triceps, measuring 46 × 17 mm, with a mixed content including blood. The lesion seemed encapsulated, with some antero-external irregularities (Figure 1). At pre-operative angiography, intense peripheral vascularization was observed, without any apparent infiltration. Furthermore, PET scan showed areas of incremented radiopharmaceutical accumulation in the belly of the left brachial triceps muscle (SUV max 6.5). It was therefore decided to perform a first cytological aspiration with a fine needle, the results of which were constituted by blood material comprising numerous pigmented histiocyte/macrophage cells. Three months later, a biopsy was performed at the level of the lesion in question which, however, was not conclusive for any diagnosis. Following this report, it was decided to perform surgical removal of the formation and the finding was evaluated by the histopathologist.

**Figure 2 diagnostics-12-02616-f002:**
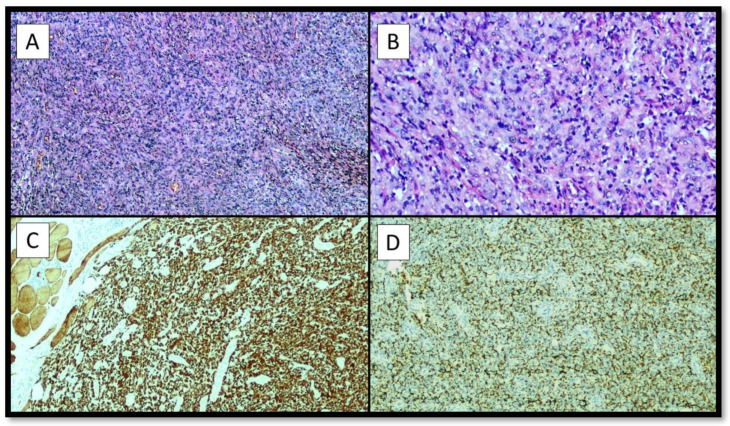
The histopathological analysis allowed to appreciate a lesion characterized by a peripheral fibrous pseudocapsule, not continuous, and by a large central cystic-haemorrhagic portion, consisting of medium-sized cellular elements (**A**), partly epitheliomorphs, sometimes with clarified cytoplasm (**B**), and in some areas spindle, mostly monomorphic, with evident eosinophilic nucleoli, with solid growth (**B**). A rich inflammatory infiltrate was described, also consisting of foamy histiocytes and hemosiderophages. The mitotic count was 2/10 high power field (HPF) and there were no areas of necrosis. The immunohistochemical reactions were positive for Desmin with a score of +++ (**C**), CD68 PG-M1 positive with a score of ++/− (**D**). Reactions for Epithelial Membrane Antigen (EMA), S-100 protein, Miogenin and MyoD1 and CD34 were negative. The fraction of neoplastic proliferation evaluated by Ki67 + was low, around 5–6%. The diagnosis of Angiomatoid Fibrous Histiocytoma (AFH) was then made and the patient was closely followed-up. After less than 6 months, the patient was again subjected to a surgical resection, whose histological sample was analyzed and found to be an intramuscular localization of AFH with free margins. After about 12 months, the patient returned to the referring physician with progressive febrile episodes and asthenia, as well as mild breathing difficulties. A new CT was performed which showed a thickening of about 0.8 cm. An apical resection of the upper lobe of the right lung (LSD) was performed.

**Figure 3 diagnostics-12-02616-f003:**
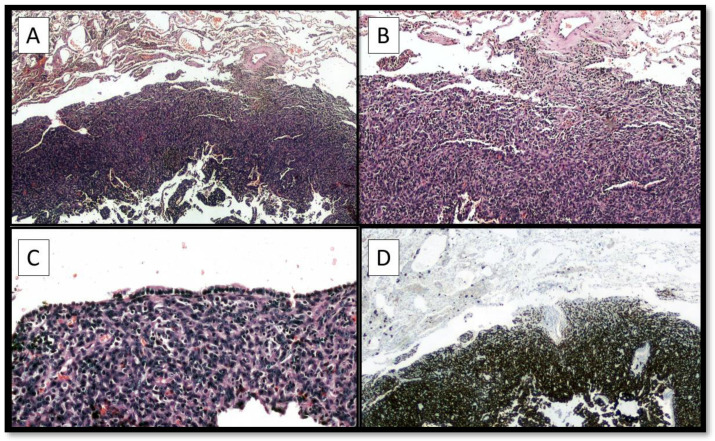
(**A**,**B**) The histological analysis highlighted a lesion with morphological characteristics very similar to the previous AFH, although the organ location had changed. In (**C**) was possible to appreciate the respiratory mucosa that covers a part of AFH. Immunohistochemical investigations were again conducted which were positive for Desmin (**D**), CD68, with a Ki67 + of about 10–15%. A diagnosis of metastasis in the upper lobe of the right lung of AFH was made and the sample was sent for FISH investigation for diagnostic confirmation. Molecular analyzes revealed the presence of the EWSR1::CREB1 fusion gene. This feature was very important to confirm and start the therapeutic path [1,2,3]. Regarding the differential diagnosis, it is important to differentiate AFH from other types of dermatofibroma, or, in the case of prominent myxoid change, from other myxoid tumors such as low-grade fibromyxoid sarcoma, extraskeletal myxoid chondrosarcoma and myxoid liposarcoma [4]. The case presented is of particular interest for three reasons: (1) AFH metastasis rate described in the literature is around 1–5%, with about 15 cases of AFH lung involvement described [5,6]; (2) from the few published cases of metastatic AFH, it would seem that the presence of the EWSR1::CREB1 fusion gene predisposes more aggressive behavior of the neoplasm [7]; careful follow-up with the possibility of performing a FISH in cases of relapsed AFH is mandatory [1,2,8,9,10,11,12].

## Data Availability

Not applicable.

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
