# Peer review of "Angiomatoid Fibrous Histiocytoma (AFH) of the Right Arm: An Exceptional Case with Pulmonary Metastasis and Confirmatory EWSR1::CREB1 Translocation"

_diagnostics, 2022, doi:10.3390/diagnostics12112616_

Round 1

Reviewer 1 Report

The authors Cazzato et al report a rare entity of angiomatoid fibrous histiocytoma affecting the right arm of a 62-year-old man. The patient relapsed 3 years after first surgical excision. Next, after a further 3 years the patient presented an intramuscular localization and after 1 year pulmonary metastasis. Fish analysis detected EWSR1-CREB1 gene rearrangement.

The paper is interesting and has a good relevant to the field.

The authors should address the following points:

The authors should stress the important role of NGS-based analysis a supporting tool for diagnosis. In this regard the following paper should be included in the references:

·         Racanelli, D. et al. Next-generation sequencing approaches for the identifcation of pathognomonic fusion transcripts in sarcomas: The experience of the Italian ACC Sarcoma Working Group. Front. Oncol. 10, 489. https://doi.org/10.3389/fonc.2020.00489 (2020).

If possible the authors should report the patient management after the occurrence of pulmonary metastasis in terms of chemotherapy, radiotherapy etc. In this regard the authors should underline the role of chemotherapy including trabectedin in the management of patients with advanced diseases. The following paper should be included.

·         Update on the role of trabectedin in the treatment of intractable soft tissue sarcomas. Onco Targets Ther. 2017 Feb 23;10:1155-1164. doi: 10.2147/OTT.S127955. PMID: 28260930; PMCID: PMC5328291.

Furthermore the following papers should be included for proprer discussion:

·         Fibrous Histiocytoma: A Tumor With Uncertain Behavior and Various Clinicopathological Presentations. Cureus. 2022 Sep 9;14(9):e28985. doi: 10.7759/cureus.28985. PMID: 36225497; PMCID: PMC9541999.

·         Durable response to crizotinib in metastatic angiomatoid fibrous histiocytoma with EWSR1-CREB1 fusion and ALK overexpression. Ann Oncol. 2022 Aug;33(8):848-850. doi: 10.1016/j.annonc.2022.05.003. Epub 2022 May 12. PMID: 35568279.

·         Molecular diagnosis of an atypical case of angiomatoid fibrous histiocytoma based on detection of the EWSR1 gene translocation. J Dermatol. 2021 May;48(5):e215-e216. doi: 10.1111/1346-8138.15823. Epub 2021 Feb 23. PMID: 33624342.

Finally the protocol number approved by Ethic Committee for that study should be included in the manuscript.

Minor revisions are requested.

Author Response

Dear Reviewer n'1,

first of all thank you very much for your congratulations that we very appreciated.

We add all references that you kindly suggest us, but we fail to have informations about therapeutic program because the patient go away in other countries (USA).

Furthermore, we have attach the informed consent because for one case report in retrospective way, we haven't protocol number.

A warm greeting, 

the authors

Reviewer 2 Report

There are some comments.

It would be better to add images showing peripheral fibrous capsule and FISH results.

It would be better to add immunohistochemical results for myogenin and/or MYOD1.

It would be better to describe differential diagnoses of angiomatoid fibrous histiocytoma.

Please edit the references according to the rules of Diagnostics.

It would be better to change as follows:

  EWSR1-CREB1 fusion -> EWSR1::CREB1 fusion

  An atypical resection -> An apical resection

Author Response

Reviewer n’2: There are some comments.

It would be better to add images showing peripheral fibrous capsule and FISH results.

Answer n’1: Dear Reviewer n’2, thank you for your valuable comments important to improve the quality of our manuscript. In the Figure 2C, in immunohistochemistry, it seems possible to appreciate the fibrous capsule of the AFH. We have not added other images so as not to make the paper too heavy. If, on the other hand, he does not agree, we will add them. Furthermore, we reported FISH analysis results without picture because the investigation was made in other institution for the lack of the probe. We have the FISH results that we reported. A warm greeting

Reviewer n’2: It would be better to add immunohistochemical results for myogenin and/or MYOD1.

Answer n’2: Thank you very much. We have the immunohistochemical figures of Miogenin and Myo D1 but, as they are consistently negative, we have not added them. We added these informations in the paper.

Reviewer n’2: It would be better to describe differential diagnoses of angiomatoid fibrous histiocytoma.

Answer n’3: all done. Thanks again.

Reviewer n’2: Please edit the references according to the rules of Diagnostics. It would be better to change as follows: EWSR1-CREB1 fusion -> EWSR1::CREB1 fusion; An atypical resection -> An apical resection.

Answer n’4: Done, thank you for all.